# Structures and Anti-Allergic Activities of Natural Products from Marine Organisms

**DOI:** 10.3390/md21030152

**Published:** 2023-02-25

**Authors:** Na Chen, Shanshan Zhang, Ansar Javeed, Cuiqin Jian, Yi Liu, Jinlyu Sun, Shandong Wu, Peng Fu, Bingnan Han

**Affiliations:** 1Zhejiang Key Laboratory of Silkworm Bioreactor and Biomedicine, Laboratory of Antiallergic Functional Molecules, College of Life Sciences and Medicine, Zhejiang Sci-Tech University, Hangzhou 310018, China; 2Hangzhou Zheda Dixun Biological Gene Engineering Co., Ltd., Hangzhou 310018, China; 3Department of Allergy, Beijing Key Laboratory of Precision Medicine for Diagnosis and Treatment of Allergic Diseases, National Clinical Research Center for Dermatologic and Immunologic Diseases, Peking Union Medical College Hospital, Chinese Academy of Medical Sciences, Peking Union Medical College, Beijing 100730, China; 4Key Laboratory of Marine Drugs, Ministry of Education of China, School of Medicine and Pharmacy, Ocean University of China, Qingdao 266003, China

**Keywords:** anti-allergic, secondary metabolites, marine organisms, molecular docking

## Abstract

In recent years, allergic diseases have occurred frequently, affecting more than 20% of the global population. The current first-line treatment of anti-allergic drugs mainly includes topical corticosteroids, as well as adjuvant treatment of antihistamine drugs, which have adverse side effects and drug resistance after long-term use. Therefore, it is essential to find alternative anti-allergic agents from natural products. High pressure, low temperature, and low/lack of light lead to highly functionalized and diverse functional natural products in the marine environment. This review summarizes the information on anti-allergic secondary metabolites with a variety of chemical structures such as polyphenols, alkaloids, terpenoids, steroids, and peptides, obtained mainly from fungi, bacteria, macroalgae, sponges, mollusks, and fish. Molecular docking simulation is applied by MOE to further reveal the potential mechanism for some representative marine anti-allergic natural products to target the H1 receptor. This review may not only provide insight into information about the structures and anti-allergic activities of natural products from marine organisms but also provides a valuable reference for marine natural products with immunomodulatory activities.

## 1. Introduction

Allergic diseases are considered to be major chronic diseases globally [1]. There are several types of allergic diseases, some of which are more common, such as allergic asthma, allergic rhinitis, anaphylactic shock, hay fever, and dermatitis [2]. Pollen allergy was first described in 1870, before that the understanding of allergic diseases was limited. Asthma in children began to increase in 1960 [3]. By 1990, asthma rose to an epidemic level due to increased sensitivity to indoor allergens, reduced diet and physical activity, and long-term shallow breathing [4]. Since 1990, food allergy has risen significantly and has now reached epidemic numbers [5]. British studies revealed that the prevalence of eczema, hay fever, and health care may be stabilizing or even declining, but the incidence of systemic allergic diseases continues to rise [6]. Allergic diseases have an increased burden on people [7]. Environmental changes such as urbanization, industrialization [8], and continuous changes in lifestyles are considered to be reasons for the raised prevalence of different forms of allergic diseases [9]. However, the current role of anti-allergic drugs mainly includes the adjuvant treatment of corticosteroids and antihistamines [10]. These treatments will bring many side effects, like drowsiness, dryness of mouth, decreased vision, and inattentiveness [11]. Developing natural products for the treatment of allergies has become increasingly important because of the increasing demand.

About 71 percent of the surface of the earth is covered by sea, which is a complex ecosystem. It has been more than four billion years since the most primitive life appeared in oceans [12]. Over billions of years of life’s evolution, a marine world of rich biodiversity has emerged, comprising primarily of marine plants, marine animals, and marine microorganisms [13]. The low similarity of living environments between marine and terrestrial organisms has resulted in the production of a variety of metabolites. The special environment of high salt and pressure, low temperature, oligotrophicity, hypoxia, and limited light determines that the secondary metabolites of marine organisms have very unique traits compared with secondary metabolites of terrestrial organisms [14]. Secondary metabolites of marine natural products possess many biological effects like anti-tumor, anti-inflammatory, anti-allergic, antiviral, antibacterial, etc. [15].

Since the 20th century, people have paid much attention to marine-derived bioactive secondary metabolites [16]. A large number of works of literature have reported anti-food allergy activity and isolated many compounds from marine organisms [17,18]. According to different chemical structures, isolated marine natural products can be divided into terpenoids, steroids, aromatic compounds, non-isoprene compounds, alkaloids, etc. Modern techniques will encourage people to extract more compounds with unique structures and biological properties from nature through the innovation of natural exploration methods [19]. Therefore, a series of studies on marine anti-allergic active molecules has been carried out worldwide. This review focuses on the structure, source, and anti-allergic activity of bioactive compounds derived from marine organisms in recent years, which provides a valuable reference for marine natural products with immunomodulatory activity.

## 2. Material and Methodology for Literature Survey

The review contains original research articles published in the English language from 1992 to 2022. During the year 2022, different database searches were performed on PubMed, Web of Science, MDPI, Elsevier, and Springer Link using keywords such as “ anti-allergy” + “marine organisms“, ”anti-allergy” + “mangrove plants“, ”anti-allergy” + “marine algae “, “anti-allergy + sea corals“, “anti-allergy” + “marine microorganisms“, and combined keyword searches. We only included studies that were completely consistent with the subject of this review. The purpose of this review article is to summarize the recent research on antiallergic compounds obtained from marine plants, animals, and microorganisms, and to provide enough information relevant to the topic of this review article. In order to assist readers, we have divided marine organisms into three parts: marine plants, marine animals, and marine microorganisms, to describe the anti-allergic activity of various compounds produced by them.

## 3. Chemical Structures and Biological Properties of the Anti-Allergy Compounds Linked with Marine Plants, Marine Microorganisms, and Marine Animals

### 3.1. Marine Plants

#### 3.1.1. Natural Products Derived from Marine Plants with Anti-Allergic Activity

In the study of anti-allergy, the most studied marine plants are algae, along with some mangrove plants. Major kinds of algae are red algae, green algae, and brown algae, which are mostly found in intertidal and subtidal zones [20]. Asian countries are rich in algal resources, especially China, Japan, and Korea. Since ancient times, humans have known of marine algae and are using it for different purposes. The secondary metabolites found in it are abundant, because they are rich in proteins, fatty acids, minerals and vitamins, and are often used as food. Some algal species also have great potential as cosmetics, drugs, and drug adjuvants [21], which play an important role to treat fever, cough, dermatitis, allergies, and other diseases [22]. Humans and other biological organisms consume a large amount of polyphenols, which is the largest compound group present in plants [23]. Polyphenols have a wide range of functions like antioxidant capacity, scavenging free radicals, and metal-chelating activity, and it is beneficial to human health, and can be used to treat and prevent cancer, cardiovascular disease, and other pathology [24]. In marine algae, the structure of most anti-allergic natural products is polyphenols. Three compounds were isolated from *Ecklonia cava* by Li et al. [25] which stimulated human basophilic KU812F cells with IgE and anti-IgE antibodies. At 100 μM, the relative levels of histamine released by three compounds **1, 2,** and **3** (Figure 1) were 23.97, 44.26 and 34.54%, respectively. Then calcium ionophore A23187 was used to mediate the degranulation of KU812F cells and RBL-2H3 cells. Three compounds at 100 μM inhibited histamine release from both cells. After flow cytometry analysis, it was proved that three compounds play an anti-allergic role by inhibiting FcεRI and IgE binding activity, and the inhibition rates were 30.58, 47.60 and 34.23%, respectively. Compounds **1** and **3** had the strongest inhibitory effect on histamine release, with IC_50_ values of 31.65 μM (RBL-2H3), 44.20 μM (KU812F), 38.87 μM (RBL-2H3), and 65.81 μM (KU812F). These compounds inhibited allergic reactions in a dose-dependent manner. Han et al. [26] also studied the anti-allergic effect of eckol (compound **2**) separated from *Ecklonia cava* (brown algae) through BMCMC (mouse bone marrow-derived mast cells) stimulated by bovine serum albumin (BSA)/immunoglobulin E (IgE) and allergic reaction models. The results showed for the first time that the compound **2** inhibited mast cell activation by inducing degranulation and cytokine production after IgE/BSA exposure. 100 μg/mL of compound **2** remarkably decreased the release of β-hexosaminidase by inhibiting the production of Th2 cytokines, i.e., IL-5, IL-4, IL-13. It reduced FcεRI expression on the cell surface, and compound **2** binds to the active site of IgE for blocking the IgE binding to FcεRI. Two bioactive phloroglucinol derivatives DHE (compound **4**) and PFF-α (compound **5**) were isolated from *Ecklonia stolonifera* (brown algae) by Shim et al. [27], as shown in Figure 1. They studied the effects of these two compounds on human basophilic KU812F cells and found that compounds **4** and **5** inhibited FcεRI expression on the surface of the cell by 16.9 and 15.4% at 50 μM, respectively. At the same time, these two compounds lessened the expression of total FcεRI α chain protein and mRNA in a dose-dependent manner, and inhibited the increase of intracellular Ca^2+^ stimulated by CRA-1, thereby exerting anti-allergic effects. Vo et al. [28] studied the Fucofuroeckol-A (F-A/compound **6**, see Figure 1) protective effect obtained from *Ecklonia stolonifera* on UVB-induced RBL-2H3 mast cell allergic reaction. They found that 50 μM F-A inhibited the fusion of granules and plasma membrane by inhibiting the increase of calcium ion concentration in mast cells, thereby inhibiting degranulation of mast cells and reducing histamine release from mast cells (release level 31%). Sugiura et al. [29] isolated and purified six compounds **2**, **3**, **7**–**10** (Figure 1) from *Eisenia arborea*, and studied the effect of oral administration of these compounds compared with EGCG, which is a known natural product with anti-allergic activity [30]. Their studies have shown that these compounds exhibited anti-allergic activity by inhibiting the release of chemical mediators like leukotriene B4, prostaglandin E2, and histamine, as well as inhibiting the cyclooxygenase-2 (COX-2) mRNA expression. The inhibitory activity of **3**, **8**, and **9** was the strongest, and the anti-allergic effect was equal to or higher than EGCG. Matsui et al. [31] isolated three compounds from *Sargassum carpophyllum*, and found that compounds **11**, **12,** and **13** (Figure 1) inhibited the release of prostaglandin D2, tumor necrosis factor-α and β-hexosaminidase in RBL-2H3 cells stimulated by DNP-HSA, with a value of 50.7, 35.9 and 43.5 μM IC_50_ values. β-hexosaminidase release was employed as an indicator of mast cell degranulation. At 40 μM, all three compounds significantly inhibited ROS production, and compound **13** slightly reduced the level of Ca^2+^.

Other compounds of brown algae also have good anti-allergic effects. Onodera et al. [32] compared Peridinin **14** and fucoxanthin **15** (Figure 1) isolated from *Symbiodinium sp* and *Petalonia fascia,* respectively. They found that topical application was better, and that compound **14** better inhibited delayed-type hypersensitivity compared to compound **15**. Compound **14** may be a potential drug for inhibiting allergic inflammation by inhibiting the migration of ear eosinophils to eotaxin and the production of eotaxin.

Chen et al. [33] studied fucoidan derived from *Cladosiphon okamuranus*. They discovered that local application of fucoidan to mice might increase Treg cell development and secrete transforming growth factor-1, which would inhibit Th2 cell-mediated immunity. Additionally, fucoidan decreased serum IgE levels and memory B cell numbers, alleviating the signs of allergic dermatitis in mice models caused by DNCB. They also investigated the effects of fucoidan derived from *Cladosiphon okamuranus* [34] on atopic dermatitis in vitro and in vivo, and discovered that this fucoidan effectively inhibited degranulation of P815 mast cells treated with C48/80 by impeding histamine and IL-4, IL-13 production. In addition, they also found that fucoidan showed comparable therapeutic effects to corticosteroids, and no side effects of corticosteroids were found in in vivo experiments. Anti-allergic activity of alginate, which was extracted from brown algae, Laminaria japonica, was studied by Yu et al. [35], and they found that after alginate treatment, the serum IgE and histamine decreased significantly in OVA-induced mice and inhibition of mast cell degranulation. Th1 and Th2 cells released IFN-γand IL-4, respectively. Alginate blocked Th0 cells’ development into Th2 cells produced by OVA and achieved an anti-allergic effect by regulating T cell population balance as a result of a significant drop in IL-4 level and a significant increase in IFN-γlevel. Alginate additionally raised the Treg cell quantity in the OVA-induced mouse spleen tissue. Liu et al. [36] detected the anti-allergic effects of R-phycocyanin (RPC) in antigen-sensitized mice and mast cells after isolating it from *Porphyra haitanensis*. Using RPC, the researchers found that the mast cell allergic reactions could be reduced through significant reductions in tropomyosin (TM)-specific IgE, along with a reduction in histamine release and mast cell degranulation. It suppresses the secretion and production of cytokines IL-4 and IL-13 and thus inhibiting the conversion of natural T cells to Th2 cells, which alleviates allergic symptoms. Vo et al. [37] extracted two peptides LDAVNR (P1) and MMLDF (P2) from microalgae (Spirulina maxima), which were investigated for their anti-allergic effects. The P1 and P2 treatments did not induce cytotoxicity and inhibited degranulation of mast cells by inhibiting the release of histamine, and the increase of intracellular calcium and cytokine production by inhibiting calcium and microtubule-dependent signaling pathways, which is the mechanism responsible for P1’s inhibitory effects. Meanwhile, P2 inhibition leads to the production of reactive oxygen species and to phospholipase Cγ activation thereby inhibiting degranulation.

#### 3.1.2. Crude Extracts from Marine Plants as Potential Sources with Anti-Allergic Activity

Kim et al. [38] treated ovalbumin (OVA)-sensitized mice with Ecklonia cava (EC) extracts, and found that EC extracts significantly inhibited allergic responses before the last airway OVA challenge. IL-4, IL-5, and Th2 cytokines play an imperative role in the instigation of allergic response. Han et al. [39] studied the anti-allergic effects of ethanolic extract of copper algae on passive cutaneous anaphylaxis and IgE/BSA-mediated mouse bone marrow activation of mast cells. Studies showed that the extract of copper algae (SHE) inhibited the β-hexosaminidase and histamine release, and substantially inhibited the degranulation of bone marrow mesenchymal stem cells. In addition, flow cytometry analysis showed that SHE markedly decreased the FcεRI binding to IgE and FcεRI expression on the surface of BMCMCs, and regulated the expression levels of mRNA of cytokines and chemokines in IgE/BSA-stimulated BMCMCs, thereby improving activation of mast cells stimulated by immunoglobulin E/bovine serum albumin. Herath et al. [40] studied whether *Sargassum horneri* ethanol extract (SHE) attenuated the effects of atmospheric particulate matter (PM) exposure on asthma. By lowering mRNA levels of the transcription factors STAT5 and GATA3, they discovered that copper algae blocked Th2 polarization and decreased the expression of IL-4, IL-5, IL-13, and Th2 cytokines in lung tissue homogenates of mice with asthma caused by PM. Additionally, oral administration of SHE dramatically decreased mast cell activation, serum IgE levels, and PM-aggravated Th2 and Th17 responses in asthmatic mice. Compounds with potential anti-allergic activity are present in red algae in addition to the structure and chemical characteristics of the anti-allergic natural products in brown algae, which have also been thoroughly investigated. Jung et al. [41] used a 95% ethanol extraction method to extract *Laurencia undulata* (LU) and showed that it contains an enormous quantity of polyphenols, and observed its anti-asthmatic effect on ovalbumin (OVA)-induced allergic respiratory reactions in mice. The results showed that LU administered prior to the last airway OVA challenge significantly inhibited allergic reactions. Shi et al. [42] investigated the anti-allergic effects of sulfated polysaccharide from *Porphyra haitanensis* (PHPS) administered orally on mice that were allergic to tropomyosin (TM). The findings of that experiment revealed that PHPS can stimulate the Treg/Th1 cytokines production like IL-10 and interferon-γ in the presence or absence of allergens. Han et al. [43] applied RASP (red algae sulfated polysaccharide) to effervescent tablets for anti-allergy research, and it was acquired by extraction of *Gracilaria lemaneiformis* and *Porphyra haitanensis*. As a result of RASP treatment, serum IgE levels, mast cell protease-1, and histamine were reduced. RASP treatment can reduce IL-4, significantly increases IFN-γ, and IFN-γ as Th1 cytokines, and promotes Th1 cell differentiation, thereby regulating allergic reactions caused by Th1/Th2 immune response imbalance. The natural products found in microalgae also have anti-allergic properties. Additionally, anti-allergic compounds have also been found in green algae. Raman et al. [44] observed that the crude extract of *Enteromorpha compressa* reduced the level of IgE induced by food allergens such as ovalbumin and that it enhanced immune function by decreasing plasma cell generation of IgE antibodies against food allergens. Cryptomonas, another algal species, has also been shown to have anti-allergic properties. The effect of *Polyopes affinis* ethanol extract on Th2-mediated allergen-induced airway inflammation in an asthmatic mouse model was evaluated by Lee et al. [45]. Researchers found that continual intraperitoneal injection of *P. affinis* ethanol extract before the last respiratory OVA challenge significantly inhibited the response and reduced ovalbumin-specific IgE by 72%.

Mangrove is a wetland woody plant community composed of evergreen trees or shrubs mainly composed of mangrove plants growing in the intertidal zone of tropical and subtropical coasts. Acharyya et al. [46] studied the anti-allergic activity of the Ethanol extract of *Lumnitzera racemosa* and the polyphenols related to this activity (**16**–**24**, see Figure 1). Oral administration of the ethanol extract of *L. racemosa* significantly reduced the number of sneezes, scratches, and nasal pain, as well as the number of lymphocytes, neutrophils, and eosinophils, and significantly inhibited TDI-induced allergic symptoms. (See Table 1 for details on compounds).

### 3.2. Marine Animals

#### 3.2.1. Natural Products Derived from Marine Animals with Anti-Allergic Activity

In the study of anti-allergy, marine animals mainly include sponges, mollusks, sea cucumbers, corals, etc. A variety of marine animals, including sponges, mollusks, and fish also have anti-allergic properties. The sponge was the first multicellular animal, living in the ocean 600 million years ago, with a high capacity for filtration [47]. In mollusks, sea cucumbers and abalone are the main sources of anti-allergy compounds. Ko et al. [48] investigated the passive cutaneous anaphylaxis of gastrointestinal digestive components of the intestinal digestive digest of abalone *Haliotis discus hannai* and a bioactive peptide (compound **25,** see Figure 2) was isolated from the gastrointestinal digestion. Histamine release could be reduced by 300μg/mL of compound **25**. Mice treated with compound **25** showed significant inhibition of the immunoglobulin E-mediated PCA response. By regulating PMA + A23187, compound **25** stimulates HMC-1 cells to produce tumor necrosis factor-α, IL-1, and IL-6 reduces the release of histamine and has anti-allergic activity.

Jiao et al. [49] identified anti-allergic terpenoids isolated from the marine sponge *Dysidea villosa* and they found that four compounds, **26**–**29** (Figure 2), suppressed the release of degranulation marker β-hexosaminidase with IC_50_ values of 8.2, 10.2, 19.9 and 16.2 μM, respectively, in a dose-dependent manner. As a result of antigen stimulation, the production of LTB 4 and IL-4 in RBL-2H3 mast cells was dose-dependently inhibited. Compound **26** demonstrated the greatest anti-allergic activity out of the four compounds. In some studies it has been shown that mast cell activation is inhibited by compound **26** by inhibiting the signaling pathway of Syk/PLCγ-1, thereby inhibiting mast cell degranulation and down-regulating LTB 4 and IL-4 production. Hong et al. [50] isolated three compounds (including **30**–**32**, see Figure 2) from the South China Sea sponge *Hippospongia lachne* to find that they inhibited IgE-stimulated RBL-2H3 cells from releasing β-hexosaminidase. It was found that compounds **30** and **31** had higher β-aminocaproic glycosidase inhibitory activity. LTB4 production by activated RBL-2H3 cells was significantly inhibited by compounds **30** and **31** with IC_50_ values of 49.37 and 23.91 μM, respectively. Andrew et al. [51] found that the marine sponge *Petrosia sp.* contained a sterol-like compound called IZP-94005 (Compound **33** as shown in Figure 2). Both in vivo and in vitro allergic reactions were studied using ovalbumin-induced bronchoconstriction and smooth muscle contractions. Based on a concentration-dependent inhibition of OVA-stimulated sensitized tracheal ring response, IZP-94005 had an IC_50_ of 10 μM. A substantial lowering in histamine release was observed after the application of IZP-94005. Shoji et al. [52] isolated two new triterpenoids with 14 carboxyl groups from the Okinawan marine sponge *Penares incrustans*. Anti-IgE-induced histamine release from rat peritoneal mast cells was inhibited by compounds **34** and **35** (Figure 2) with IC_50_ values of 1.5 μM and 10 μM, respectively. It was found that compound **34** was 17 times more potent in nature than disodium cromoglycerate (DSCG). Takei et al. [53] characterized the Okinawan marine sponge *Xestospongia bergquistia* and isolated different terpenoids from it. Dose-dependent inhibition of the release of histamine from mast cells in male Wistar rats was observed with compounds **36** and **37** (Figure 2). Release of histamine from IgE-activated mast cells was blocked by compounds **36** and **37** at 100 μM each. PI-PLC activity and inhibition of IP3 production were initiated by compound **36** in a dose-dependent manner. Aside from inhibiting calcium mobilization in intracellular calcium stores, compound **36** also inhibited calcium influx. Isolation of two terpenoids from the Okinawan marine sponge *Penares incrustans* was also performed by Takei et al. [54]. It was shown that compounds **38** and **39** (Figure 2) inhibited anti-IgE-induced histamine release in Wistar rats. At 100 μM, the anti-IgE-induced histamine release was inhibited at 90.7 ± 2.3%, 0.5 and 1.5 μM IC_50,_ respectively. There was a dose-dependent inhibition of PLA2 (phospholipase A2) activity with both compounds. This system was able to measure the IC_50_ values for PLA2 activity at 2 and 0.1 μM, respectively. (See Table 1 for details on compounds).

Pozharitskaya et al. [55] isolated and studied the anti-allergic effect of compounds (**40**–**43**, see Figure 2) of green sea urchin shell pigment. Green sea urchin shell pigment compounds had a dose-dependent inhibitory effect on histamine-induced ileum contraction in guinea pigs, ID_50_ = 1.2μg/mL. The inhibitory effect on the ocular allergic inflammation model was better than that of the reference drug olopatadine.

Most of the compounds isolated from soft corals belong to terpenoids, which mainly have cytotoxicity and anti-tumor activity, especially lactone diterpenoids, while compounds with anti-allergic activity account for a minority [56]. Shoji et al. [57] isolated four compounds (**44a**–**44d**, see Figure 2) from the soft coral *Sinularia abrupta*. Compounds **44a**–**44d** inhibited anti-IgE-induced histamine release from rat peritoneal mast cells in a dose-dependent manner. The IC_50_ values of **44a**–**44d** were 0.04, 0.6, 1.5, and 0.2 μM, respectively. It is 6500 times more potent than the well-known anti-allergic drug sodium cromoglycate (IC_50_ = 262 μM).

#### 3.2.2. Crude Extracts from Marine Animals as Potential Sources with Anti-Allergic Activity

A research study by Kim et al. [58] examined the ability of oral administration of LMW-AV (low molecular weight peptides) acquired from gastrointestinal digestion of Abalone viscera (AV) to treat (AD) atopic dermatitis in a dermatitis-induced model stimulated with Dermatophagoides farinae. In AD-like lesions, LMW-AV inhibited the expression of chemokines and cytokines related to Th2, and it inhibited serum IgE levels. Eosinophils were decreased as a result of oral LMW-AV treatment, skin thickness was reduced, mast cell infiltration into the epidermis was inhibited, and skin edema was reduced.

Lee et al. [59] investigated the anti-allergic activity of sea cucumber and demonstrated that the liquid salting-out extract of sea cucumber activated and recruited regulatory T and Treg cells that improved allergic airway inflammation. Moreover, sea cucumber extract rich in palmitoleic acid inhibited IgE better than extracts poor in palmitoleic acid, whereas palmitoleic acid lowers serum total immunoglobulin E (IgE) concentrations.

Fish have a rich diversity due to their complex living environment. There is a variety of biological activities associated with different parts of fish. Willemsen [60] found that fish oil has an effect on decreasing allergic symptoms when high n-3LCPUFA intake is coupled with low n-6PUFA intake, whereas TH2 and TH1 reactions are reduced by N-3LCPUFA (fish oil), Treg frequency increases, and IgE level is reduced, which indicates that this oil has the potential for anti-allergic activity. Aryani et al. [61] examined the anti-allergic properties of charcoal from the inedible part of *Channa pleurophthalmus Blkr*, (Kerandang fish), and anti-hyaluronidase activity was determined by anti-hyaluronidase test. Based on the results, caudal fin charcoal extract exhibited the highest inhibitory effect and pectoral fin charcoal extract exhibited the lowest inhibitory effect. With four mg/mL, ethyl acetate extract concentration of caudal fin charcoal showed the greatest inhibitory effect on hyaluronidase. A potential anti-allergic drug can be developed from its non-edible parts.

### 3.3. Marine Microorganisms

#### Natural Products Derived from Marine Microorganisms with Anti-Allergic Activity

Harunari et al. [62] studied the activity of Hyaluromycin **45** (Figure 3), a new member of the rubromycin family isolated from marine-derived *Streptomyces* sp., which is composed of γ-rubromycin core structure with 2-amino-3-hydroxycyclopent-2-enone (C5N) unit as amide substituent of the carboxyl group. The enzyme hyaluromycin imparts a major role in allergic responses and in mast cell degranulation. We found that hyaluromycin had a 25-fold higher inhibitory effect against hyaluronidase than the plant terpenoid glycyrrhizic acid with 14 μM IC_50_ value, therefore providing new insights in the development of anti-allergic drugs. Niu et al. [63] isolated a polyketone compound **46** (Figure 3) from a deep-sea-derived fungus *Graphostroma sp.* and tested its biological activity in IgE-mediated rat basophilic leukemia-2H3 cells. Compound **46** can also be isolated from the fermentation broth of Streptomyces sp. The findings showed that compound **46** significantly inhibited histamine release and degranulation in RBL-2H3 cells, with a 13.7 μM IC_50_ value. It was found that the methyl group present at C-3, the C-6 hydroxyl group, and the methoxy group at C-7 were essential for anti-food allergy activity. They also isolated eight tetracyclic diterpenoids from the deep-sea fungus *Botryotinia fuckeliana* in the western *Pacific* [64]. Compound **47** (Figure 3) was found with a novel 6/6/5/5 tetracyclic carbon skeleton. Compared with loratadine (positive control and IC_50_ = 0.1 mM), compound **47** showed anti-allergic effects in RBL-2H3 cells (IC_50_ = 0.2 mM).

Shu et al. [65] isolated an alkaloid **48** (Figure 3) from *Penicillium*, i.e., deep-sea fungus. Their study revealed that compound **48** significantly reduced β-hexose release and histamine in RBL-2H3 cells induced by ovalbumin (OVA) in a dose-dependent manner (IC_50_ = 6.67 μg/ml), and it had no cytotoxic effect on RBL-2H3. There was a dose-dependent decrease in mast cell protease-1, histamine, immunoglobulin E, and tumor necrosis factor-α levels, and an increase in IL-10 production. The increase of calcium ions is the key process of MC secretory granule translocation. Compound **48** significantly inhibited the accumulation of calcium ions in RBL-2H3 cells in a dose-dependent manner, thereby blocking the activation of macrophages and inhibiting mast cell degranulation.

Uras et al. [66] purified butyrolactone I (compound **49,** see Figure 3) from *Aspergillus terreus*. Inhibition of calcium ion carrier A23187 and antigen-induced degranulation is manifested by its significant anti-allergic activity, with 39.7 and 41.6 μM, IC_50_ values. Elsbaey et al. [67] isolated two compounds (**50**, **51,** see Figure 3) from the white bean culture of the endophytic fungus *Aspergillus amstelodami*. Anti-allergic activity of quercetin was determined in 100 μM RBL-2H3. Both compounds significantly reduced the release of β-hexosaminidase and had no significant cytotoxicity to cells. These compounds may have some anti-allergic effects, although they have a lower efficacy than quercetin. Xie et al. [68] isolated a new cyclic ether compound nesterenkoniane (**52**) and 12 known compounds from *Nesterenkonia flava*, an actinomycete originating from the deep sea (see Figure 3). By employing IgE-mediated rat mast cell RBL-2H3 as a model, cyclo-(D)-proline-(D)-leucine (compound **53,** see Figure 3) and indole-3-carbaldehyde (compound **54,** see Figure 3) showed significant anti-allergic activity with 69.95 and 57.12 μg/mL IC_50_ values, respectively. (See Table 1 for details on compounds).

**Table 1 marinedrugs-21-00152-t001:** Research Overview of Marine Natural Products with Anti-allergy Activities. (See Appendix A for further details.).

Source of Compounds	The Sources of Isolation	Number of Compounds	Range of Dosage	Structure Type	Test System	Targets/Pathway/Process Mechanism	Reference
Marine Plants	*Ecklonia cava*	Compound **1**–**3**	100 μM	Polyphenol	Human basophilic KU812F cells and RBL-2H3 cells	FcεRI and IgE binding activity, histamine release, degranulation of cell	[25]
	*Ecklonia stolonifera*	Compound **4**–**5**	50 μM	Polyphenol	Human basophilic KU812F cells	The expression of FcεRI, intracellular Ca^2+^	[27]
	*Ecklonia stolonifera Okamura*	Compound **6**	50 μM	Polyphenol	RBL-2H3 mast cell	Ca^2+^ concentration, mast cell degranulation, histamine release	[28]
	*Eisenia arborea*	Compound **2**,**3**,**7**–**10**	10–200 µM	Polyphenol	DNP-BSA-induced RBL-2H3 mast cell	Release of histamine, leukotriene B4 and prostaglandin E2, H_1_ receptor	[29]
	*Sargassum carpophyllum*	Compound **11**–**13**	40 μM	Polyphenol	DNP-HSA-induced RBL-2H3 cells	Release of β-hexosaminidase, mast cell degranulation	[31]
	*Symbiodinium* sp., *Petalonia fascia*	Compound **14**–**15**	50 μg	Carotenoid	BALB/cAJc1 mice	Migration of eosinophils	[32]
	*Lumnitzera racemosa*	Compound **16**–**24**	/	(Ethanol extract)	Toluene 2,4-diisocyanate (TDI)-induced allergic model mice	IgE	[46]
Marine Animals	*Haliotis discus hannai*	Compound **25**	50 mg/kg	Polypeptide	Passive cutaneous anaphylaxis in mice	Histamine release, FcεRI and IgE binding activity	[48]
	*Sponge*	Compound **26**–**29**	250 μg/mL	Terpenoids	RBL-2H3 mast cells	β-hexosaminidase, Syk/PLCγ-1, mast cell degranulation	[49]
	*Hippospongia lachne*	Compound **30**–**32**	200 μg/mL	(Ethanol extract)	IgE-stimulated RBL-2H3 cells	β-hexosaminidase	[50]
	*Petrosia* sp.	Compound **33**	3–30 μM	Sterol	OVA-induced mice	Histamine release levels	[51]
	*Penares incrustans*	Compound **34**–**35**	0–10 μM	Triterpenoids	Anti-IgE-induced mast cells	Histamine release	[52]
	*Xestospongia bergquistia,* *Penares incrustans*	Compound **36**–**39**	100 μM	Terpenoids	Anti-IgE-induced male Wistar rats’ mast cells	IP3 production, Histamine release, intracellular Ca2+, PLA2	[53,54]
	*Green sea urchin*	Compound **40**–**43**	1.2 μg/mL	Polyhydroxy-1,4-naphthoquinone	Histamine-induced guinea pigs	β-hexosaminidase	[55]
	*Sinularia abrupta*	Compound **44**	0.04–1.5 μM	Polyhydroxysteroid	Anti-IgE-induced mice	Mast cell, histamine release	[57]
*Marine Microorganisms*	*Streptomyces* sp.	Compound **45**	/	Macrolide	/	Mast cell degranulation, hyaluronidase	[62]
	*Graphostroma sp.* *Botryotinia fuckeliana*	Compound **46**–**47**	0–200 μM	Tetracyclic diterpenoids	RBL-2H3 cells	Histamine release, mast cell degranulation	[63,64]
	*Penicillium*	Compound **48**	20 mg/kg	Quinoline alkaloid	OVA-induced RBL-2H3 cells	β-hexose and histamine, mast cell degranulation, IgE	[65]
	*Aspergillus terreus*	Compound **49**	100 μM	Hemiterpenes	RBL-2H3 cells	β-hexosaminidase, IgE	[66]
	*Aspergillus amstelodami*	Compound **50**–**51**	100 μM	β-lactams, adenine	RBL-2H3 cells	β-hexosaminidase	[67]
	*Nesterenkonia flava*	Compound **52**–**54**	1.0–80.0 µg/mL	Cycloethers, diketopiperazine, alkaloid	RBL-2H3 cells	IgE, β-hexosaminidase	[68]

## 4. Potential Mechanism Study of Representative Natural Products

Allergens of various types can cause food allergies, like peanuts [69], cockroaches, and dust mites in household dust [70], as well as certain drugs like sulfonamides, penicillins, and certain vaccines. According to research reports, the main secondary metabolites extracted from marine organisms have an effect on allergic reactions which include: the inhibition of Th2 cells to secrete allergy-related cytokines and chemokines; inhibition of the binding of IgE to FcεRI receptor; inhibition of the release of histamine to inhibit mast cell degranulation (Figure 4).

Once the allergen enters the body, the antigen-representing cells receive it, leading to the transformation of helper cells from Th0 cells to Th2 cells [71]. Different cytokines are secreted by Th2 cells, including interleukin 4,5,13, which cause activation of B cells to release IgE [72,73], and cross-link with the FcεRI receptor expressed on mast cells and basophils surface, triggering biochemical reactions, such as FcεRI-dependent signal cascade activation, increased intracellular calcium levels, microtubule polymerization and degranulation [74]. Subsequent histamine, interleukin, protease, chemokines, prostaglandins, and other inflammatory mediators are released in large quantities, leading to a variety of allergic reactions [75]. For regulating type I allergies and treating allergic diseases, these allergic cascades are considered molecular targets.

Previous studies have shown that whether it is marine plants, animals or microorganisms, or synthetic anti-allergic marine secondary metabolites, their inhibitory mechanisms on allergic reactions are similar, mainly including inhibition of FcεRI and IgE binding activity, histamine release, mast cell degranulation, and cytokine production. In addition, it also controls allergic reactions by inhibiting the expression of FcεRI on the cell surface, inhibiting the flow of Ca^2+^, and regulating the balance of Th1 cells and Th2 cells. The above studies have shown that there is a link between the release of chemical mediators and mast cell degranulation [26].

Mast cells are granulocytes [76] that are widely distributed around microvessels in the skin and visceral mucosa. They contain heparin, histamine, and 5-hydroxytryptamine. Due to contact between IgE antibodies and antigens bound to mast cells, the cells are mostly collapsed, and particles and substances are released by the disintegration of the cells, which can cause rapid allergic reactions in tissues. At the same time, they secrete a variety of cytokines and participate in immune regulation. It is mainly the result of antigen-induced aggregation of FceRI receptor molecules on the surface of mast cells, which accounts for the triggering of the release of inflammatory mediators by mast cells. The inflammatory mediators released by mast cell activation initiate a signal transduction cascade.

In mast cells, the initial factor of Ca^2+^ influx is the cross-linking of allergen and the IgE-FcεRI complex to activate phospholipase C (PLC), which can produce phosphatidylinositol 4,5-diphosphate (PIP2). Secondary messengers are formed from PIP2, such as IP3 (inositol 1,4,5-triphosphate) and DAG (diacylglycerol). IP3 combines on the endoplasmic reticulum membrane with the IP3 receptor (IP3R) to release Ca^2+^ from the endoplasmic reticulum Ca^2+^ store [77]. Ca^2+^ influx is primarily caused by endoplasmic reticulum Ca^2+^ stores depletion [78,79]. Its molecular mechanism is that there are proteins STIM1 and STIM2 containing EF-hand domains in the endoplasmic reticulum, which can sense Ca^2+^ depletion in the calcium store of the endoplasmic reticulum [80]. Then it migrates to the plasma membrane, interacts with the Orai protein on the plasma membrane, and opens the Ca^2+^ channel on the cell membrane to allow the influx of extracellular Ca^2+^ [81]. As shown in Figure 5, Ca2 + participates in mast cell activation signaling pathway.

Mast cells are characterized by a higher content of electron-dense secretory granules filled with numerous pre-activated immunomodulatory compounds which are significant effector cells of the immune system [87]. When mast cells are activated, they first undergo a process of degranulation. These preformed granular compounds are released quickly into the surrounding environment [88]. Studies have shown that in the presence of IL-4 and free IgE, the FcRI expression on the mast cell’s surface increases, which further enhances the activation process [89]. This shows that IL-4 is not only imperative for the Th2 cells’ differentiation during allergic sensitization, but also for the differentiation of Th2 cells during allergic stimulation [90]. Granular material secreted by mast cells is initially synthesized in the Trans-Golgi apparatus. First, the small vesicle propagules bud on the Trans-Golgi and then undergo massive fusion to form immature mast cell granules with minimal dense centers. Next, the dense centers of immature mast cell granules need to grow for maturation, a process that is accomplished by the filling of granules and compounds like cytokines, proteases, and bioactive amines.

The research has pointed out that the increase of intracellular Ca^2+^ level triggers cell degranulation, which makes the granules move from the interior of the cell to the cytoplasmic membrane. Subsequently, coronin 1A and coronin 1B regulate cortical actin depolymerization [91]. The premise of degranulation is a large number of particle fusion, which is regulated by a large number of soluble N-Ethylmaleimide-sensitive factor attachment protein receptors, including target-SNARE and vesicle-SNARE proteins [92]. Subsequently, a degranulation reaction occurs and pre-formed particulates are released in a soluble form, while other compounds remain in the matrix, known as particulate residues. In this matrix, proteoglycans are the most important components, including tumor necrosis factor, carboxypeptidase A3, and chymase [93]. Mast cell granule maturation is largely dependent on the participation of proteoglycans. Mast cells also release granule inclusions through exocytosis, a process accomplished by vesicle germination on the granule [94]. Current research on the molecular mechanism of mast cell degranulation mainly focuses on tyrosine kinase Lyn and Fyn-dependent signal transduction pathway and increase of intracellular Ca^2+^ level [95]. There are many other signaling pathways involved to regulate mast cell degranulation, deserving further exploration, which can provide a scientific basis for determining the network system between various signals in mast cells and provide a further reference for elucidating the mechanism of allergy.

Mast cells are activated by an allergen that binds to serum IgE attached to their FcεRI receptors, they release cytokines, eicosanoids, and their secretory granules. When the same allergen appears again, the cross-links cell surface to IgE and FcεRI activates mast cells through signal transduction and releases active mediators in the granules, triggering type I hypersensitivity. Therefore, blocking the binding of FcεRI and IgE effectively inhibits allergic reactions. Most natural products extracted from marine organisms have this activity. Compounds can also reduce the binding of FcεRI and IgE by inhibiting the expression of FcεRI on the surface of mast cells.

In short, the key anti-allergic target is to inhibit mast cell degranulation, so the production of allergic mediators is inhibited. In an allergic reaction, marine natural products can prevent allergic reactions in the following way. Firstly, they inhibit the upstream pathway by inhibiting the production of IgE, reducing the binding of IgE to FcεRI, inhibiting the Th2 cytokines release, and regulating the balance of Th1 and Th2 cells; then they inhibit the binding of IgE to FcεRI and inhibit the expression of FcεRI on the mast cells surface; and finally, the downstream part of the pathway inhibits mast cell degranulation and inhibits the release of inflammatory factors such as histamine. At present, a variety of histamine receptor antagonists have been developed, and only a few are in clinical trials. The disadvantage is that the half-life is too short [96]. Most receptor antagonists have binding activity with receptors [97] and show good results in the treatment of a variety of allergic diseases. In addition, with the use of anti-allergic drugs in the pathway of allergic reactions, some receptors will experience side effects, such as antihistamine drugs leading to human inattention, lethargy, arrhythmia, etc. The development of antagonists for different allergic inflammatory diseases remains a major challenge.

## 5. Discussion and Future Prospects

At present, the first-line treatment of anti-allergic drugs mainly includes the adjuvant treatment of topical glucocorticoids and anti-histamines [98]. Long-term use will produce adverse side effects and drug resistance [99]. For example, hormonal drugs can cause side effects like obesity and swelling in patients. Antihistamines can cross the blood-brain barrier, so after administration people feel dizziness, inattentiveness, etc. In order to explore how to reduce side effects and drug resistance, adopt new targets and mechanisms, and develop drugs with good efficacy and few side effects, this manuscript reviews anti-allergic natural products extracted from marine organisms to find the best way to treat allergic diseases. Marine-derived anti-allergic natural products have great potential. Many researchers have been employing various techniques to discover and synthesize novel compounds to improve the diversity and availability of marine compounds. There are thousands of organisms in the marine environment, ranging from single-cell organisms to mammals. Each organism carries unique substances and has rich drug development potential. At the same time, allergies affect generations of people, therefore advancement in research of anti-allergic drugs is very important.

The increasing demand for new drugs from natural sources has promoted the expansion of modern biotechnology research to find alternative sources of bioactive components with potential applications in various fields. A large number of active metabolites are present in the marine environment with diverse chemical structures. In recent years, many researchers have shown great interest in in vivo and in vitro experiments for exploring the effects of marine-derived compounds as anti-allergic drugs. Anti-allergic secondary metabolites are employed against many allergic reactions through different pathways. Most of the compounds with anti-allergic activity are derivatives of the polyphenol structure obtained from marine brown algae. Many biological activities, such as antioxidant [100], antibacterial, anti-inflammatory [101], anti-cancer [102], anti-virus, etc. can be achieved by these types of compounds. Studies have shown that polyphenols also have a certain inhibitory effect on allergic reactions. In marine animals, anti-allergic compounds i.e., terpenoids are mostly derived from sponges. In addition to being a rich reservoir of microbial diversity, the sponge is also one of the largest contributors to the overall diversity of marine microorganisms on the planet [103]. The special structure of the sponge leads to the establishment of complex microbial symbiosis [104]. In this article, molecular docking MOE was used to explore the mechanism of some secondary metabolites extracted from marine organisms affecting mast cell degranulation, that is, histamine release. ASP107 is a conserved residue in amine receptors that forms an anchor salt bridge with the ligand amine moiety. It has been documented that this interaction is necessary for H_1_R antagonists and agonists binding in mutation studies. Molecular docking studies were performed on 18 reported marine anti-allergic natural products acting on H1R receptors. According to the interaction between scoring and key amino acids, compounds **14** and **37** were selected for display. The molecular docking results are shown in Figure 6. It is proved that the phenolic hydroxyl group of compound **14** forms a hydrogen bond interaction with the key amino acid Asp107. The docking energy is −8.2685 Kcal/mol. The phenolic hydroxyl group of compound **37** formed a hydrogen bond interaction with the key amino acid Asp107, and the docking energy was −5.6156 Kcal/Mol. The docking fraction of H1R with the standard ligand cetirizine (−7.3Kcal/mol) showed a satisfactory docking fraction [45]. Compared with the reported drugs, in addition to the key amino acids, the phenolic hydroxyl group of compound **14** forms hydrogen bond interactions with Thr132 and Tyr431, and the amide oxygen of compound **37** also has hydrogen bond interactions with Asn198, further indicating that it has good affinity with H_1_R and is expected to be a candidate drug for H_1_R antagonists with high selectivity. However, there is still a lack of experiments to verify the targets of marine natural products. (The method of molecular docking is shown in Appendix A)

Finally, we hope that this review will not only provide information on the anti-allergic activity and structure of marine natural products but also provide a valuable reference for how marine natural products inhibit allergic reactions as well as immunomodulatory effects. At present, there are relatively few studies compiled on these anti-allergic marine natural products, but many secondary metabolites with good biological activity have been discovered in recent years. Further exploration is undoubtedly a useful strategy to discover new anti-allergic drugs.

## Figures and Tables

**Figure 1 marinedrugs-21-00152-f001:**
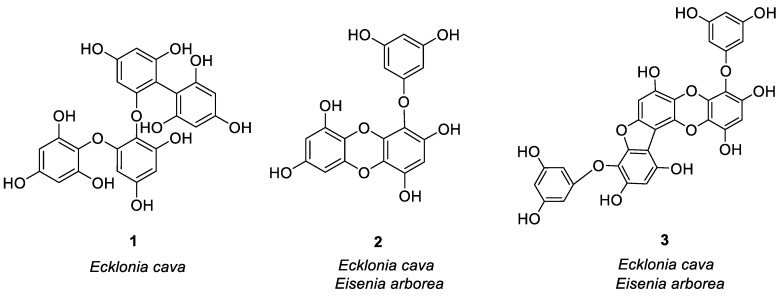
Structures of compounds **1**–**24**.

**Figure 2 marinedrugs-21-00152-f002:**
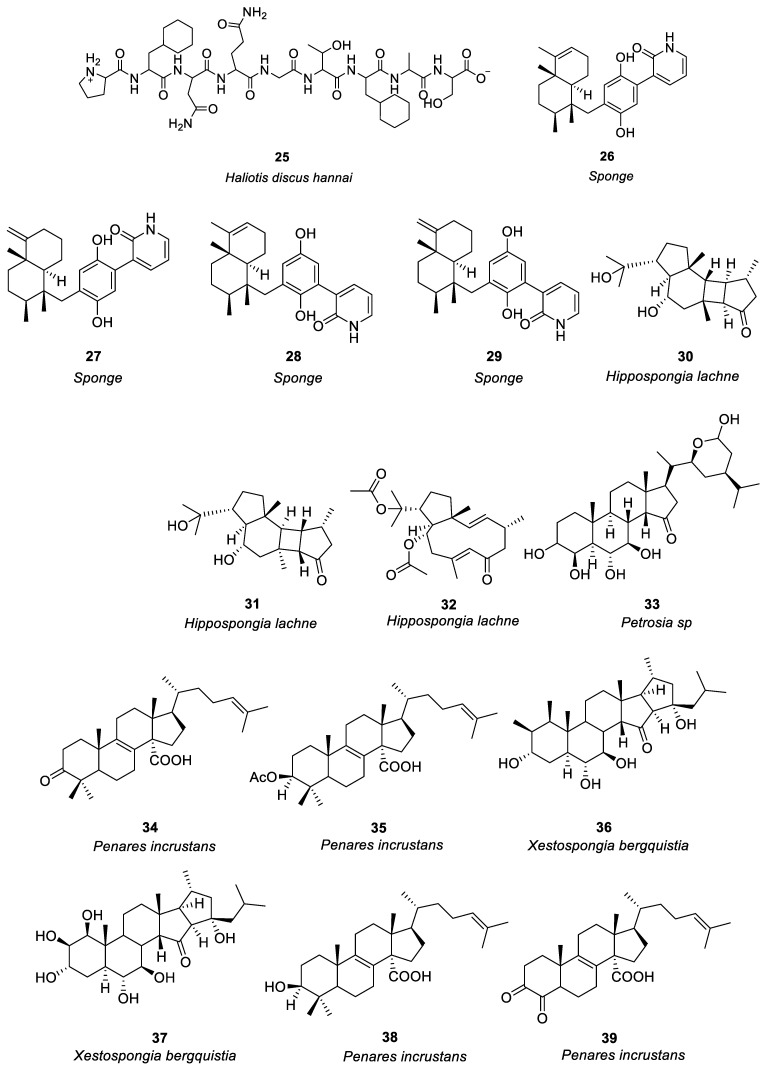
Structures of compounds **25**–**44**.

**Figure 3 marinedrugs-21-00152-f003:**
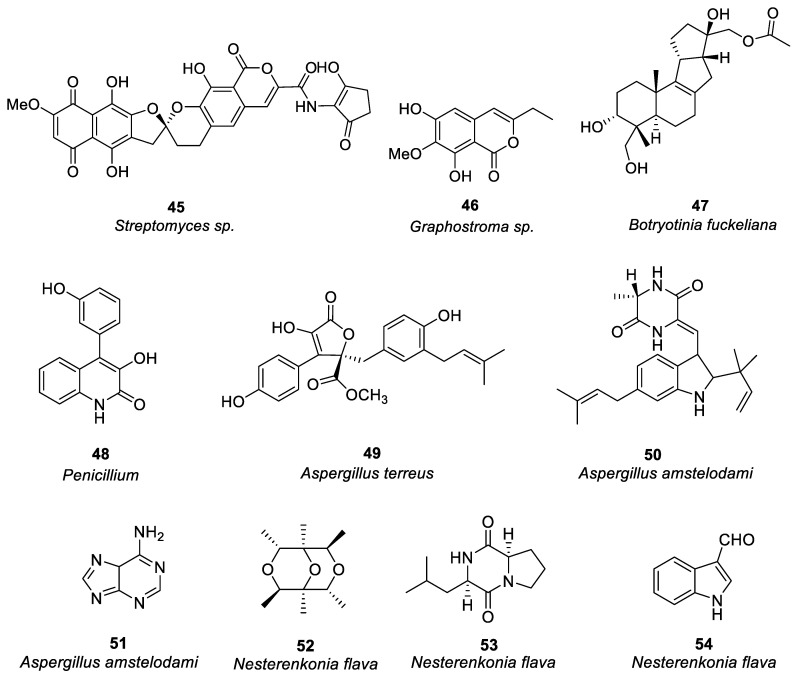
Structures of compounds **45**–**54**.

**Figure 4 marinedrugs-21-00152-f004:**
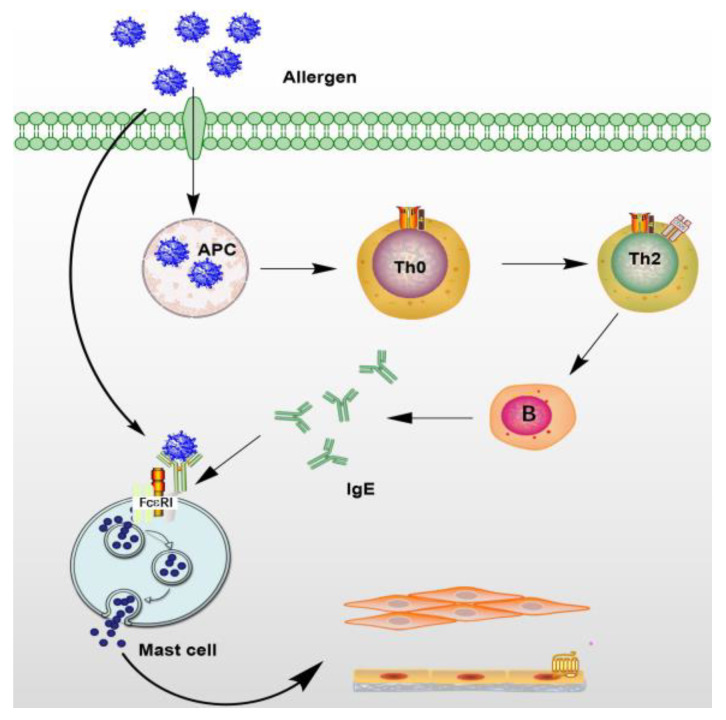
Anti-allergic targets of marine natural products.

**Figure 5 marinedrugs-21-00152-f005:**
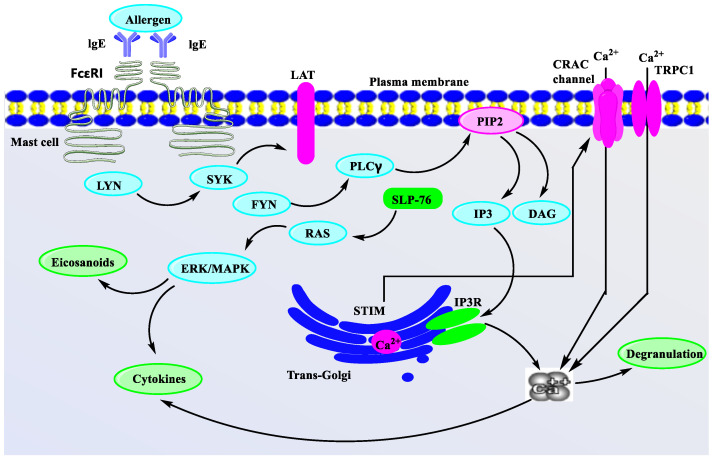
Ca^2+^ participates in mast cell activation signaling pathway. Intracellular Ca^2+^ regulates cell granule migration, granule membrane fusion, and cell degranulation [82]. This figure shows only FcεRI (high-affinity IgE receptor) on inactivated mast cells followed by the activation of FcεRI cross-linking, and the linker for activation of T cells. LAT is phosphorylated in a manner depending on tyrosine-protein kinase Lyn and Syk [83]. Degranulation was accompanied by activation of signal transduction phospholipases PLCγ, protein kinase C and increased calcium ions, and TRPC1 channels further facilitated Ca^2+^ influx [84]. RAS-RAF-MAPK pathway activation leads to eicosanoid production (including leukotrienes C4 and prostaglandin D2) and cytokines. InsP3R (Inositol Triphosphate Receptor), a membrane glycoprotein complex activated by InsP3, acts as a calcium channel [85] and STIM (Stromal interaction molecules) is a calcium receptor on the endoplasmic reticulum [86].

**Figure 6 marinedrugs-21-00152-f006:**
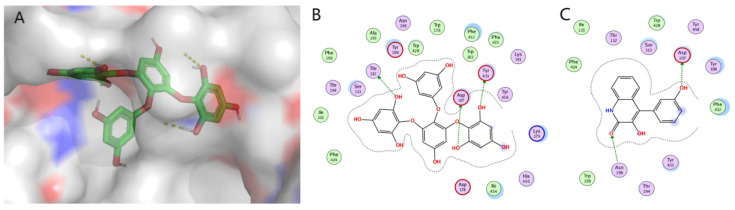
Molecular docking of compounds 14 (**A**: 3D, **B**: 2D) and 37 (**C**: 2D) binding to histamine H_1_ receptor [105]. (Molecular docking simulation was performed on MOE, and the visualization software was Pymol).

## Data Availability

Not applicable.

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
