# Peer review of "Structures and Anti-Allergic Activities of Natural Products from Marine Organisms"

_marinedrugs, 2023, doi:10.3390/md21030152_

Round 1

Reviewer 1 Report

In this submitted manuscript, authors wanted to give an overview of marine natural compounds with anti-allergic activities. These compounds were categorized on the basis of their origin organisms. Subsequently, they tried to analyze the potential mechanism for the anti-allergic compounds. The novelty of the topic of this manuscript is very good, and this manuscript fits the scope of Marine Drugs absolutely. Moreover, the authors did quite a lot work on this topic, especially the extensive literature research as revealed by more than one hundred citations of references.

However, I will recommend it for acceptance after major revision.

Comments:
1. What were the definition of ‘marine plants’ and ‘marine animals’? As shown in the manuscript, only algae were reviewed for section 2.1. How about the mangroves? Were there no compounds with anti-allergic activities reported? And corals, sea serpent and other animals should be considered.
2. The compounds should be recorded with detailed information, including their names and numbers. However, the names of very few compounds were provided. Compounds were not numbered sequentially in the text. For instance, compound 19 was mentioned after 33. And every compound mentioned should be numbered. The compound numbers 7, 8 and 27 were missing in the text. Please check it throughout the whole manuscript.
3. According to the content of section 3.4, authors wanted to give examples of the anti-allergic derivatives of marine natural products deduced from chemical synthesis or modification. However, only one examples was insufficient.
4. For section 3, it is better to analysis the reported targets/pathway/process mechanism of those reviewed compounds, including the advantages and/or disadvantages, which could give an insight of the application prospects or the fundament for development to drug leads.
5. What’s the purpose of molecular docking for compounds 19, 14 and 37? To find a target or confirm the reported target? As mentioned in the text, ‘compounds 13, 14 and 15 inhibited the release of prostaglandin D2, tumor necrosis factor-α and β-hexosaminidase in RBL-2H3 cells stimulated by DNP-HSA in a dose-related manner’, but the molecular docking focused on the binding modes between 14 and H1 receptor. Please keep in mind that this manuscript is a review, it is not proper to record the author's research work in it.

Others:
1. Please adjust the reference list to Marine Drugs style.
2. Table caption should be put before the table body.
3. Please check the grammar and typo errors throughout the whole manuscript, such as ‘compounds 44’ in the Figure 4 caption.

Reviewer 2 Report

I have read the manuscript and I have some questions and comments.
1. The manuscript does not contain a "Materials and Methods" section. Therefore, it is not clear how it differs from many others published recently on a similar topic (for example, https://doi.org/10.1007/s11101-018-9547-3, DOI: 10.1039/C8RA04777D, https://doi. org/10.3390/md20110693 and others). Please include a "Materials and Methods" section indicating the purpose of this manuscript, years of search, keywords, databases, etc.
2. Please structure Figure 1 by indicating the source from which the compound was isolated and the literature reference.
3. The manuscript is lacking of important details. Without these details, it seems that the authors collected information not from full articles, but from abstracts. Sections 2.2, 2.3, please supplement with a table indicating the name of the compound, the source of its isolation, test system, positive control, data in numbers, reference. Discuss the data.
3. Previously, molecular docking into H1R receptor structures obtained from molecular dynamic simulations showed that all spinochrome derivatives bind to the receptor active site, but spinochrome monomers fit better to it (DOI: 10.1055/s-0033-1351098). Include required information and compare data for monomeric and dimeric structures.
4. The list of references is formatted incorrectly (for example, 89, 91 and many others). Please correct.
5. Section "4. Discussion and future prospect" is very poor. There is no discussion. No perspective shown. This section needs to be rewritten.
6. Figures 5-8 must be supplemented with information about their literary sources.
7. In Table 1, please include the structures of the compounds and their names.

Reviewer 3 Report

Line 22-23           Not a true statement particularly as marine derived metabolites are frequently from microbes with many not yet cultivated. Similarly terrestrial sourced compounds are often the product of epiphytes and/or endophytes. For example, terrestrial Streptomyces have over 800 terpenoid gene clusters from data in the early 2000s.

Line 65-66           A very simplistic and incorrect statement. Obviously, the authors have not kept up with the data from the Piel and associated groups and many others, Simply look at the very complex structures of the ecteinascidins and in particular the halichondrins. Both have led to highly active approved antitumor agents

Upto Line249      Section 2.1 is a mixture of results from pure compounds and extracts, and even the pure compounds are at high micromolar levels. This combination of “data” is effectively worthless as leads to anti-allergy compounds.

Line 249 onwards dealing with 2.2

What is effectively ignored is that the actual producer(s) of these compounds are almost certainly not the microorganism from which they were isolated but are the product(s) of single-celled microbes in or on the macrooganism.

Line 341 Microbes Hyaluromycin (34) bears no structural relationship to the erythromycin structures, nor is it a peptide.

The authors need to check the closest chemical relatives to any compound that they mention in this section due to their totally inaccurate comments above.

Line 386  PDI appears to be synthetic. Why is it mentioned?

Table 1 needs refining as to levels used and what would be the 50% inhibition level? 100 micromolar with a pure compound is not drug like.

Line 497  Compound 19 requires a very high level of the peptide to show a response that this is effectively a meaningless exercise. Similar problem with compound 14.

A partially marked up pdf is listed below

Round 2

Reviewer 1 Report

This manuscript has been improved by authors. However, revisions are still required.

1. The structures of some compounds shown in Figures 1 and 2 were questionable. As observed, the chemical structures of the pairs of compounds 2 and 7, 3 and 11, 16 and 7, 43 and 44 were identical. Please check them.

2. For the four polyhydroxysteroids isolated from the soft coral Sinularia abrupta, the numbers 14 assigned for them should be adjusted in the text and Figure 2.

3. There was no Figure 4 in the revised manuscript, the order of figures needs to be adjusted.

4. The text of third row should be adjusted. ‘Name of compound’ didn’t mean numbers.

5. The reference style should be adjusted.

Reviewer 2 Report

I have read the revision manuscript. I have repeated questions and comments.

1. The manuscript does not contain a "Materials and Methods" section. Therefore, it is not clear how it differs from many others published recently on a similar topic (for example, https://doi.org/10.1007/s11101-018-9547-3, DOI: 10.1039/C8RA04777D, https://doi. org/10.3390/md20110693 and others). Please include a "Materials and Methods" section indicating the purpose of this manuscript, years of search, keywords, databases, etc.

2. The manuscript is lacking of important details. Without these details, it seems that the authors collected information not from full articles, but from abstracts. Sections 2.2, 2.3, please supplement with a table indicating the name of the compound, the source of its isolation, test system, positive control, data in numbers, reference. Discuss the data.3. Figures 5-8 must be supplemented with information about their literary sources. Please complete the figure captions with literary reference.

4. The heading of table 1 is incorrect. The table does not contain treatment information. These are not drugs. Please correct the title of table 1.

Reviewer 3 Report

This is still a random compilation of data. What needs to be done to rescue the reasonable parts of the review is to separate the "compound mixtures" that are not identified to a single pure compound and use the pure compounds to show the activities/diseases that are relevant as the major part of the review. Then one can take the section(s) dealing with undefined to partially purified compounds/sources and discuss those as "potential sources once pure compound(s) are isolated and identified". Simply listing crude and pure compounds under the same heading is a worthless exercise.

Round 3

Reviewer 2 Report

I have read the revision manuscript. The authors have made the necessary corrections and I have no more questions.